# Lightspeed Black-box Bayesian Optimization via Local Score Matching

**Yakun Wang**
University of Bristol
yakun.wang@bristol.ac.uk

**Sherman Khoo**
University of Bristol
sherman.khoo@bristol.ac.uk

**Song Liu**
University of Bristol
song.liu@bristol.ac.uk

## Abstract

Bayesian Optimization (BO) is a powerful tool for tackling optimization problems involving limited black-box function evaluations. However, it suffers from high computational complexity and struggles to scale efficiently on high-dimensional problems when fitting a Gaussian process surrogate model. We address these issues by proposing a fast acquisition function maximization procedure. We leverage the fact that Probability Improvement (PI) acquisition function can be seen as a likelihood function whose score can be estimated through a simple linear regression problem called local score matching. This enables fast gradient-based optimization of the acquisition function, and a competitive BO procedure which performs similarly to that of computationally expensive neural networks.

## 1   Introduction

Black-box optimization [1, 2] seeks to identify the optimum (maximum in this paper) of a function $g(\mathbf{x})$ whose closed-form expression and gradient information are unknown, with as little computational resources as possible. Bayesian optimization (BO, [6]) is particularly effective for this task. It utilizes a probabilistic surrogate, typically a Gaussian Process (GP, [20]), to sequentially select new evaluation points based on the mean function and quantify uncertainty through the covariance function. Although BO can achieve relatively good accuracy with only a few function evaluations, the cost of each new proposal evaluation scales as $\mathcal{O}(n^3)$ with the number of function evaluations $n$, which becomes the dominant factor for overall cost in the optimization process.

To reduce computational expenses, a new paradigm called *Bayesian optimization via density ratio estimation* (BORE, [4, 16]) was introduced. BORE reformulates the improvement-based acquisition function (e.g. probability improvement, PI[10], expected improvement, EI[11]) as a problem of estimating the density ratio [15, 22] between two distributions of $\mathbf{x}$ conditioned on whether the corresponding function value $y$ exceeds a certain threshold $\tau$. Song et al. [13] further generalize BORE to likelihood-free Bayesian optimization (LFBO), where the acquisition function can be constructed in terms of more complex utility functions through variational representation[12]. This approach significantly reduces the computational complexity from $\mathcal{O}(n^3)$ to $\mathcal{O}(n)$. However, the acquisition function itself in both BORE and LFBO still remains demanding to optimize due to:

- The involvement of neural networks typically incurs extra cumbersome numerical optimization.

- Potential overfitting in the density ratio estimate, particularly in high dimensions.

Workshop on Bayesian Decision-making and Uncertainty, 38th Conference on Neural Information Processing Systems (NeurIPS 2024).

Motivated by these challenges, we present a new approach that maximizes the acquisition function by gradient ascent. The key observation is that the PI acquisition function is a likelihood function whose gradient could be estimated using score matching. Under mild regularity conditions, we utilize a simple regression model to learn the score without training any neural network. Moreover, our method maintains the computational complexity at $\mathcal{O}(n)$.

## 2 Backgroud

Consider the global optimization problem for a black-box function $f$ over a compact search space $\mathcal{X} \subset \mathbb{R}^d$:

$$\mathbf{x}^* = \arg\max_{\mathbf{x} \in \mathcal{X}} f(\mathbf{x}), \tag{1}$$

where the expression and gradient of the objective function $f : \mathcal{X} \to \mathbb{R}$ are unknown. We can only access $f$ through a set of noisy observations $\mathcal{D}_N = \{(x_n, y_n)\}_{n=1}^N$, where the evaluations $y = g(f(\mathbf{x}); \epsilon)$ are corrupted by noise $\epsilon$.

By assigning a probabilistic surrogate model (typically analytical probabilities e.g. GP [20]) to the objective function $f$, the acquisition function $\alpha(\mathbf{x}; \mathcal{D}_N, \tau)$ is defined as the expected value of the utility function $u(\mathbf{x}, y, \tau)$ under the posterior predictive $p(y|\mathbf{x}, \mathcal{D}_N)$ of $f$ attained from GP regression [19]:

$$\alpha(\mathbf{x}; \tau) := \mathbb{E}_{p(y|\mathbf{x}, \mathcal{D}_N)}[u(\mathbf{x}, y, \tau)], \tag{2}$$

where $\tau$ is a hyperparameter that balances exploration and exploitation. BO selects the candidate solutions by maximizing acquisition $\mathbf{x}_{t+1} = \arg\max_{\mathbf{x}} \alpha(\mathbf{x}; \tau)$. In GP-based BO [5], most acquisition functions have closed-form solutions that are easy to optimize [21]. Nevertheless, the complexity of the GP regression is $\mathcal{O}(n^3)$ which limits scalability to large datasets, posing significant computational challenges in high-dimensional settings.

We focus on a particular utility function, the Probability of Improvement function (PI): $u(\mathbf{x}, y, \tau) := \mathbb{I}(y - \tau \geq 0)$, where $\tau$ is a threshold of the function value. By definition, the acquisition function is

$$\alpha(\mathbf{x}; \tau) = \mathbb{E}_{p(y|\mathbf{x}, \mathcal{D}_N)}[\mathbb{I}(y - \tau \geq 0)] = p(y \geq \tau|\mathbf{x}, \mathcal{D}_N).$$

This acquisition function measures the probability that a new candidate point $\mathbf{x}$ yields an observation $y$ greater than the threshold $\tau$. $\tau$ is typically selected as the current maximum observed function value: $\tau = \max_n y_n$ in our observation $\mathcal{D}_N$.

## 3 Methodology

First, let $z$ denote a binary variable:

$$z := \mathbb{I}(y \geq \tau) = \begin{cases} 0, & y < \tau \\ 1, & y \geq \tau, \end{cases} \tag{3}$$

Then, the PI acquisition function can be rewritten as $\alpha(\mathbf{x}; \tau) = p(z = 1|\mathbf{x})$. Without any probabilistic surrogate model to the objective function $f$, we propose to *directly* maximize the PI acquisition function by performing *gradient ascent*:

$$\mathbf{x}_k = \mathbf{x}_{k-1} + \eta \nabla_{\mathbf{x}} \log p(z = 1|\mathbf{x})|_{\mathbf{x} = \mathbf{x}_{k-1}}. \tag{4}$$

### 3.1 Matching the score locally

We model $\nabla_{\mathbf{x}} \log p(z = 1|\mathbf{x})|_{\mathbf{x} = \mathbf{x}_{k-1}}$ using a vector-valued function $\boldsymbol{\beta}(z) : \{0, 1\} \to \mathbb{R}^d$:

$$\boldsymbol{\beta}(z) := \begin{cases} \boldsymbol{\beta}_0 \in \mathbb{R}^d, & z = 0 \\ \boldsymbol{\beta}_1 \in \mathbb{R}^d, & z = 1. \end{cases} \tag{5}$$

We leverage the score matching techniques [8, 18] to learn the score of the acquisition function $\nabla_{\mathbf{x}} \log p(z = 1|\mathbf{x})$ through a least squares regression:

$$\min_{\boldsymbol{\beta}_1} \quad J(\boldsymbol{\beta}) = \mathbb{E}_{p(z, \mathbf{x}|\mathbf{x}_{k-1})} \left[ \|\nabla_{\mathbf{x}} \log p(z|\mathbf{x}) - \boldsymbol{\beta}(z)\|^2 \right], \tag{6}$$

---

**Algorithm 1** Maximizing Acquisition Function via Gradient Ascent

---

**Require:** Generator $y = g(f(\mathbf{x}); \epsilon)$, hyperparameters $\sigma$, $\eta$, threshold $\tau$, input $\mathbf{x}_{k=0}$
         steps of gradient ascent $K$, number of samples $M$

1: **for** $k = 1, ..., K$ **do**          ▷ Perform gradient ascent $K$ steps
2:     **while** $m \leq M$ **do**          ▷ Survey local landscape
3:        sample $\mathbf{x}^{(m)} \sim q(\mathbf{x}|\mathbf{x}_{k-1})$
4:        evaluate $y^{(m)} \leftarrow g(f(\mathbf{x}); \epsilon)$
5:        $z^{(m)} \leftarrow \mathbb{I}[y^{(m)} \geq \tau]$
6:        $m \leftarrow m + 1$
7:     **end while**
8:     $\mathcal{D}'_k \leftarrow \{(\mathbf{x}^{(m)}, y^{(m)})\}_{m=1}^M$          ▷ Collect sample
9:     $\widehat{\boldsymbol{\beta}_1} \leftarrow$ Monte Carlo approximate according to (9)          ▷ Local score estimation
10:     $\mathbf{x}_k \leftarrow \mathbf{x}_{k-1} + \eta\widehat{\boldsymbol{\beta}_1}$          ▷ Update location
11: **end for**
12: Output: the final candidate $\mathbf{x}_K$, used sample $\mathcal{D}'_{1:K}$

---

where $\mathbf{x}_{k-1}$ is from the previous value of $\mathbf{x}$ in the gradient ascent algorithm. The joint probability is factorized as $p(z, \mathbf{x}|\mathbf{x}_{k-1}) = p(z|\mathbf{x})q(\mathbf{x}|\mathbf{x}_{k-1})$[1], where $q(\mathbf{x}|\mathbf{x}_{k-1})$ is a *proposal distribution*. It samples candidate $\mathbf{x}$ around $\mathbf{x}_{k-1}$.

Since the target score $\nabla_{\mathbf{x}} \log p(z = 1|\mathbf{x})$ in the objective (6) is unknown, we cannot directly minimize $J(\boldsymbol{\beta})$. The following theorem provides a closed-form solution of (6) and forms the basis of our algorithm:

**Theorem 1.** *Assuming that the probability $p(z|\boldsymbol{x})$ and the proposal distribution $q(\boldsymbol{x}|\boldsymbol{x}_{k-1})$ are differentiable. The objective (6) has the optimal solution:*

$$\boldsymbol{\beta}_1^* = \mathbb{E}_{p(\boldsymbol{x}|z=1,\boldsymbol{x}_{k-1})}\left[\nabla_{\boldsymbol{x}} \log p(z = 1|\boldsymbol{x})\right] = -\mathbb{E}_{p(\boldsymbol{x}|z=1,\boldsymbol{x}_{k-1})}\left[\nabla_x \log q(\boldsymbol{x}|\boldsymbol{x}_{k-1})\right], \quad (7)$$

*where the second equality follows from integration by parts under mild assumptions.*

Note that the minimizer $\boldsymbol{\beta}_1^*$ in Eq. (7) is a conditional expectation of the desired score $\nabla_{\mathbf{x}} \log p(z = 1|\mathbf{x})|_{\mathbf{x}=\mathbf{x}_{k-1}}$, implying that it is a biased estimator. The following result ensures that it is asymptotically unbiased given an appropriate choice of the proposal distribution.

**Corollary 1.** *Let the proposal distribution $q(\boldsymbol{x}|\boldsymbol{x}_{k-1})$ be a normal distribution $N(\boldsymbol{x}|\boldsymbol{x}_{k-1}, \sigma^2 I_d)$. Then $\boldsymbol{\beta}_1^* = \mathbb{E}_{p(\boldsymbol{x}|z=1,\boldsymbol{x}_{k-1})}\left[\sigma^{-2}(\boldsymbol{x} - \boldsymbol{x}_{k-1})\right]$ and*

$$\lim_{\sigma \to 0} \boldsymbol{\beta}_1^* = \nabla_{\boldsymbol{x}} \log p(z = 1|\boldsymbol{x})|_{\boldsymbol{x}=\boldsymbol{x}_{k-1}}. \quad (8)$$

Corollary 1 shows the solution to the objective (6) is tractable and, when $\sigma$ goes to zero, provides an asymptotically unbiased estimate of the desired score. The proofs of Theorem 1 and Corollary 1 can be found in Appendix A.

### 3.2 Gradient ascent algorithm and finite sample estimator

We summarize our gradient ascent algorithm in Algorithm 1 and defer the full BO algorithm together with the complexity analysis to Appendix B. We now detail the computation of $\widehat{\boldsymbol{\beta}_1}$ using $\mathcal{D}'_k$, i.e., paired function evaluations and their input values. According to Corollary 1, $\boldsymbol{\beta}_1^* = \int p(\mathbf{x}|z = 1, \mathbf{x}_{k-1})\left[\sigma^{-2}(\mathbf{x} - \mathbf{x}_{k-1})\right] d\mathbf{x}$. Once the threshold $\tau$ is determined, we can approximate the above integral using a Monte Carlo estimator:

$$\widehat{\boldsymbol{\beta}_1} = \begin{cases} \frac{\sum\left[\sigma^{-2}(\mathbf{x}^{(m),1} - \mathbf{x}_{k-1})\right]}{\sum_{m=1}^M z^{(m)}}, & \sum_{m=1}^M z^{(m)} > 0 \\ 0, & \sum_{m=1}^M z^{(m)} = 0 \end{cases}, \quad (9)$$

where $\mathbf{x}^{(m),1}$ is short for $\mathbf{x}^{(m)}|z^{(m)} = 1$. If $z^{(m)} = 0$ for all $1 \leq m \leq M$ we simply set $\widehat{\boldsymbol{\beta}_1} = 0$. Intuitively, $\widehat{\boldsymbol{\beta}_1}$ can be interpreted as an average of $(\mathbf{x} - \mathbf{x}_{k-1})/\sigma^2$ using *local information* that is $\{(\mathbf{x}^{(m)}, z^{(m)})\}_{m=1}^M$ sampled within a neighborhood of $\mathbf{x}_{k-1}$ with radius determined by $\sigma$.

---

[1]By our construction, $z$ and $\mathbf{x}_{k-1}$ are conditionally independent given $\mathbf{x}_{k-1}$, so $p(z|\mathbf{x}, \mathbf{x}_{k-1}) = p(z|\mathbf{x})$.

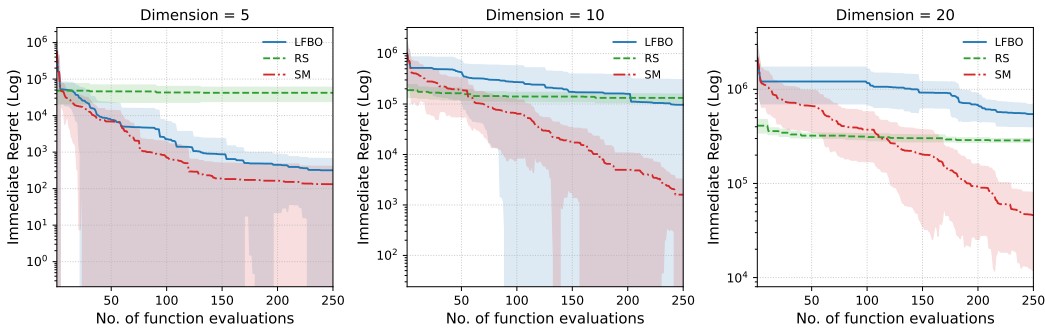

Figure 1: Immediate regret for the Rosenbrock function, repeated over 10 different seeds

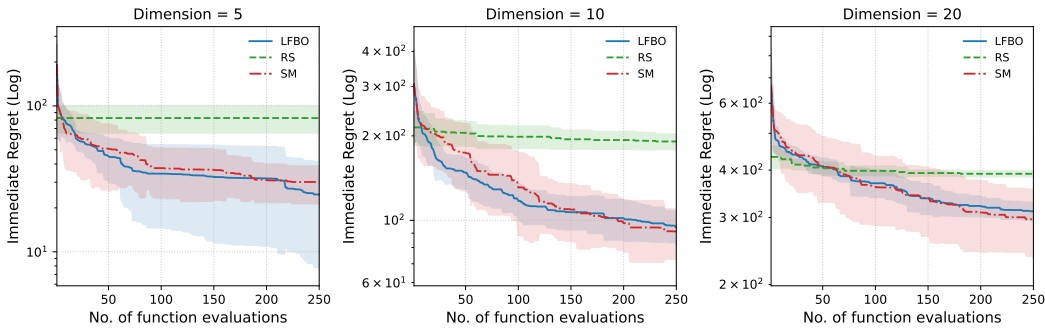

Figure 2: Immediate regret for the Rastrigin function, repeated over 10 different seeds

## 4 Experiments

Since we estimate the gradient of the acquisition function, we can resort to first-order optimizers such as ADAM [9] and RMSProp [17] to maximize the acquisition function.

Note that Corollary 1 suggests that $\sigma$ should be small near the end of gradient ascent. However, since $\sigma$ controls the local survey region as well, we propose a linear annealing schedule to balance this trade-off.

$$\sigma_t^2 = \sigma_0^2 \cdot \max\left(0, 1 - \frac{t - 0.1}{T}\right)$$

We quantitatively evaluate the performance of our method by benchmarking against some standard black-box optimization problems, fixing a function evaluation budget of 250. Following LFBO [13], we report the *immediate regret* as a metric, which measures the distance of the current best function evaluation from the optimum function evaluation. The shaded regions represent the mean plus and minus one standard deviation across 10 different seeds. We compare against the neural-network based LFBO approach, which despite using a different acquisition function, is a similar approach to our work that does not require inference of a surrogate model. We also provide the results using a random-search algorithm as a reference. From these results, we observe that our local score matching method (SM) is broadly competitive with the LFBO method. It outperforms LFBO in higher dimensions in the Rosenbrock experiment when the neural network estimation in LFBO is likely to struggle. In this work, we focus on preliminary investigation of LSM method via synthetic datasets with mild dimensions. Experiments on high-dimensional, real-world datasets are important future works.

## 5 Discussions and Future Works

We propose an optimization scheme that maximizes the PI acquisition function bypassing the GP regression restriction. Despite the promising performance of our method, there are still some limiations. As the gradient information is myopic, our algorithm may not achieve the global optimum

of the acquisition function. In addition, our method is restricted to PI acquisition function since our score matching objective can only be used to estimate the gradient of a likelihood function. Without a likelihood expression, other acquisition functions cannot be readily estimated through our procedure. An interesting avenue for future work would be to generalize our work to a broader class of acquisition functions. Finally, our method requires additional evaluations of the blackbox function for estimating the gradient of the utitlity function at each gradidient iteration.

We also notice that our algorithm is very similar to Covariance Matrix Adaptation - Evolutionary Strategy (CMA-ES, [7]). Further investigations into the relationship between these two methods could be a promising direction.

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

# A  Proofs

For ease of notation, here we denote $z_1$ and $z_0$ when $z = 1$ and $z = 0$ respectively.

## A.1  Proof of Theorem 1

*Proof.* Note that $\boldsymbol{\beta}(z)$ is a function that *only* depends on $z$. Unrolling Eq. (6) by factorizing the joint probability in two conditional density:

$$J(\boldsymbol{\beta}) = \mathbb{E}_{p(z,\mathbf{x}|\mathbf{x}_{k-1})}\left[\|\nabla_{\mathbf{x}}\log p(z|\mathbf{x}) - \boldsymbol{\beta}(z)\|^2\right] = J_1(\boldsymbol{\beta}_1) + J_0(\boldsymbol{\beta}_0) \tag{10}$$

$$= \underbrace{p(z_1|\mathbf{x}_{k-1})\int p(\mathbf{x}|z_1,\mathbf{x}_{k-1})\left[\|\nabla_{\mathbf{x}}\log p(z_1|\mathbf{x}) - \boldsymbol{\beta}_1\|^2\right]d\mathbf{x}}_{J_1(\boldsymbol{\beta}_1)} \tag{11}$$

$$+ \underbrace{p(z_0|\mathbf{x}_{k-1})\int p(\mathbf{x}|z_0,\mathbf{x}_{k-1})\left[\|\nabla_{\mathbf{x}}\log p(z_0|\mathbf{x}) - \boldsymbol{\beta}_0\|^2\right]d\mathbf{x}}_{J_0(\boldsymbol{\beta}_0)} \tag{12}$$

$$= p(z_1|\mathbf{x}_{k-1})\left[\|\boldsymbol{\beta}_1\|^2 - 2\langle\boldsymbol{\beta}_1,\nu_z(\mathbf{x})\rangle\right] + C + J_0(\boldsymbol{\beta}_0), \tag{13}$$

where $\nu_{z_1}(\mathbf{x}) = \mathbb{E}_{p(\mathbf{x}|z_1,\mathbf{x}_{k-1})}[\nabla_{\mathbf{x}}\log p(z_1|\mathbf{x})]$ and:

$$C = \mathbb{E}_{p(\mathbf{x}|z_1,\mathbf{x}_{k-1})}\left[\|\log p(z_1|\mathbf{x})\|^2\right] \tag{14}$$

is a constant. Take partial derivate w.r.t $\boldsymbol{\beta}_1$ on Eq. (13) and set it to zero, after simplifying one can recognize the optimal solution:

$$\boldsymbol{\beta}_1^* = \nu_{z_1}(\mathbf{x}) = \mathbb{E}_{p(\mathbf{x}|z_1,\mathbf{x}_{k-1})}[\nabla_{\mathbf{x}}\log p(z_1|\mathbf{x})]. \tag{15}$$

Given $p(z_1|\mathbf{x})$ and $q(\mathbf{x}|\mathbf{x}_{k-1})$ is differentiable and under mild assumption [2], one has:

$$\boldsymbol{\beta}_1^* = \mathbb{E}_{p(\mathbf{x}|z_1,\mathbf{x}_{k-1})}[\nabla_{\mathbf{x}}\log p(z_1|\mathbf{x})] \tag{16}$$

$$= \int \frac{p(z_1|\mathbf{x},\mathbf{x}_{k-1})q(\mathbf{x}|\mathbf{x}_{k-1})}{p(z_1|\mathbf{x}_{k-1})}\cdot\frac{\nabla_{\mathbf{x}}p(z_1|\mathbf{x})}{p(z_1|\mathbf{x})}d\mathbf{x} \tag{17}$$

$$\stackrel{(*)}{=} \int \frac{p(z_1|\mathbf{x})}{p(z_1|\mathbf{x}_{k-1})}\nabla_{\mathbf{x}}q(\mathbf{x}|\mathbf{x}_{k-1})d\mathbf{x} \tag{18}$$

$$= -\int \frac{p(z_1|\mathbf{x},\mathbf{x}_{k-1})q(\mathbf{x}|\mathbf{x}_{k-1})}{p(z_1|\mathbf{x}_{k-1})}\cdot\frac{\nabla_{\mathbf{x}}q(\mathbf{x}|\mathbf{x}_{k-1})}{q(\mathbf{x}|\mathbf{x}_{k-1})}d\mathbf{x} \tag{19}$$

$$= -\mathbb{E}_{p(\mathbf{x}|z_1,\mathbf{x}_{k-1})}[\nabla_{\mathbf{x}}\log q(\mathbf{x}|\mathbf{x}_{k-1})], \tag{20}$$

---

[2]Recall $z$ and $\mathbf{x}_{k-1}$ are conditionally independent given $\mathbf{x}$ and this assumption of independence $p(z = 1|\mathbf{x}) = p(z = 1|\mathbf{x},\mathbf{x}_{k-1})$ is also used in the proof of Corollary 1.

in which $(*)$ uses integration by parts with regularity condition:

$$\lim_{|x_i|\to\infty} q(\mathbf{x}|\mathbf{x}_{k-1})p(z_1|\mathbf{x}) = 0, \tag{21}$$

holds in every dimension $x_i$, $i = 1, \ldots, d$ of $\mathbf{x}$. $\qquad\square$

## A.2 Proof of Corollary 1

*Proof.* Rewrite $\boldsymbol{\beta}_1^*$ using Bayesian rule:

$$\boldsymbol{\beta}_1^* = \mathbb{E}_{p(\mathbf{x}|z_1,\mathbf{x}_{k-1})}\left[\nabla_{\mathbf{x}}\log p(z_1|\mathbf{x})\right] \tag{22}$$

$$= \int \frac{p(z_1|\mathbf{x},\mathbf{x}_{k-1})q(\mathbf{x}|\mathbf{x}_{k-1})}{p(z_1|\mathbf{x}_{k-1})}\nabla_{\mathbf{x}}\log p(z_1|\mathbf{x})d\mathbf{x}. \tag{23}$$

Now we set the proposal distribution $q(\mathbf{x}|\mathbf{x}_{k-1}) = N(\mathbf{x}|\mathbf{x}_{k-1}, \sigma^2 I_d)$ as Gaussian, whose limit distribution is is the Dirac delta function centered on $\mathbf{x}_{k-1}$ as $\sigma \to 0$. Under the same assumption in Proof of Theorem 1, we now have:

$$\boldsymbol{\beta}_1^* \to \frac{p(z_1|\mathbf{x})}{p(z_1|\mathbf{x}_{k-1})}\nabla_{\mathbf{x}}\log p(z_1|\mathbf{x})\bigg|_{\mathbf{x}=\mathbf{x}_{k-1}} \tag{24}$$

$$= \nabla_{\mathbf{x}}\log p(z_1|\mathbf{x})|_{\mathbf{x}=\mathbf{x}_{k-1}} \tag{25}$$

when $\sigma \to 0$. $\qquad\square$

# B The complete BO algorithm

---
**Algorithm 2** Bayesian Optimization via Local Score Matching

---
**Require:** Generator $y = g(\mathbf{x}; \epsilon)$, hyperparameters $\sigma$, $\eta$, budget $T, K, M$, observations $\mathcal{D}_N$
1: $\mathcal{D}_{t=0} \leftarrow \mathcal{D}_N$ (or initialize $\mathbf{x}_{t=0}$, simulate $y_{t=0} \leftarrow g(\mathbf{x}_{t=0}; \epsilon)$, $\mathcal{D}_{t=0} \leftarrow \{(\mathbf{x}_{t=0}, y_{t=0})\}$ if no observations at beginning)
2: **while** $1 \leq t \leq T$ **do**
3: $\quad\tau \leftarrow \max_y$ in $\mathcal{D}_{t-1}$ $\qquad\qquad\qquad\qquad\qquad\qquad\qquad\qquad\triangleright$ Set the threshold
4: $\quad\mathbf{x}_{k=0} \leftarrow \mathbf{x}_{t-1}$ $\qquad\qquad\qquad\qquad\qquad\qquad\qquad\qquad\triangleright$ initializing $\mathbf{x}_{k-1}$
5: $\quad$Do Algorithm 1
6: $\quad\mathbf{x}_t \leftarrow \mathbf{x}_K$ $\qquad\qquad\qquad\qquad\qquad\qquad\qquad\qquad\quad\triangleright$ Cadidate solution
7: $\quad y_t \leftarrow g(\mathbf{x}_t; \epsilon)$ $\qquad\qquad\qquad\qquad\qquad\qquad\triangleright$ Evaluate black-box function
8: $\quad\mathcal{D}_t \leftarrow \mathcal{D}_{t-1} \cup_{k=1}^{K} \mathcal{D}_k' \cup (\mathbf{x}_t, y_t)$ $\qquad\qquad\qquad\qquad\triangleright$ Update dataset
9: $\quad t \leftarrow t + 1$
10: **end while**

---

Given $T, K, M$, the total number of evaluations required in our algorithm is $n = T \times (K \times M + 1)$. The complexity is thus $\mathcal{O}(TKM)$ or equivalently $\mathcal{O}(n)$.

Theoretically, $\mathbf{x}_{t-1}$ can be initialized to any location $\mathbf{x}_{1:N}$ in $\mathcal{D}_N$. Experimentally, we set it as the $\mathbf{x}$ of which corresponding evaluation is $\max y, y \in \mathcal{D}_N$ by the greedy heuristic.

# C Experimental Details

For all algorithms, we initialized the dataset with $4$ initial points, which we do not count in our function evaluations, as is necessary for the LFBO algorithm. This initial dataset is the same across all algorithms. For the random search algorithm, the optimal point and function evaluation is initialized from this dataset.

For our local score matching algorithm, the Adam optimizer is used for the maximizing of the acquisition function. We used a fixed step-size of $5 \cdot 10^{-1}$ throughout all experiments.[3] We used a

---
[3]Inspired by the training of Noise Conditional Score Network (NCSN) [14], when the budget is considerably large we can set the learning rate $\eta \propto 1/\sigma^2$ to cancel the unstability induced by variance annealing.

budget of $T = 5, K = 5, M = 10$, which results in an overall function evaluation budget of 255, although only the first 250 function evaluations are shown in Figures 1 and 2.

For the random search algorithm, we sampled uniformly with bounds of $[-5, 5]$ across all dimensions, for both experiments. We sampled 10 points, over a total of 25 iterations, for a total of 250 function evaluations in total.

For the LFBO algorithm, we broadly followed the specifications provided by code in the original paper [13]. We used a two layer Neural Network, with 32 hidden units, which is optimized with the Adam optimizer with a fixed learning rate of $10^{-3}$. The acquisition function was optimized with the BOTORCH package [3] in Python.

