# OpenReview forum: "Lightspeed Black-box Bayesian Optimization via Local Score Matching"
_NeurIPS.cc/2024/Workshop/BDU — NeurIPS BDU Workshop 2024 Poster_

### Official Review · Reviewer_4CBb · 2024-09-28
**Review of "Lightspeed Black-box Bayesian Optimization via Local Score Matching"**

**Rating:** 6
**Confidence:** 4

**Review:**

**Summary:**
This paper proposes a new and efficient approach to optimize the Probability Improvement (PI) acquisition function for Bayesian Optimization. The key contribution is the use of local score matching to estimate the gradient of the PI acquisition function, allowing for gradient ascent without the expensive covariance matrix inversion that typically makes GP-based methods computationally intensive. The method offers computational efficiency with complexity reduced to $O(n)$, an improvement over the standard $O(n^3)$ required for GP-based methods. This paper demonstrates the efficiency of their local score matching method through experiments on synthetic objectives, comparing it against the state-of-the art $O(n)$ Likelihood-Free Bayesian Optimization (LFBO) and random search.

**Pros:**
- Reducing the complexity to $O(n)$ using local score matching is a valuable improvement over traditional GP-based methods, which typically have $O(n^3)$ complexity due to covariance matrix inversion. While there are other state-of-the-art methods such as BORE and LFBO, these approaches often rely on neural networks and suffer from challenges like overfitting in high dimensions or complex numerical optimization. The proposed method addresses these issues by using local score matching, offering a simpler and more scalable alternative.
- Applying local score matching to estimate the gradient of the PI acquisition function is a novel technique that provides a new way to optimize acquisition functions without expensive surrogate model inference.
- Providing some derivations to support the soundness of the gradient estimation via local score matching. Specifically, the authors present a closed-form solution for the score matching objective and prove that the estimator is asymptotically unbiased under certain regularity conditions.
- Validating the efficiency of their proposed method on two synthetic objective functions with dimensions ranging from 5 to 20, demonstrating its scalability in medium-high dimensions.

**Cons:** \
**Methodology:**
- The statement, "We propose to directly maximize the PI acquisition function by performing gradient ascent," could be misleading. While the approach indeed uses gradient ascent, the gradient computation is not direct but is instead estimated via local score matching. It's better to explicitly mention that "direct" refers to the difference from the standard approach, which involves GP regression to obtain the gradient.
- The paper lacks a clear explanation of why the proposed method has computational complexity $O(n)$. A brief discussion explaining the difference between the $O(n^3)$ complexity of the standard method (due to covariance matrix inversion) and the $O(n)$ complexity of the proposed method would make this more clear.
- While the paper admits that the method is restricted to the PI acquisition function, it would be useful to provide a short explanation of why this limitation exists. Specifically, this could involve discussing why the local score matching technique works well for PI but is not easily extendable to other acquisition functions like EI.

**Experiment:**
- The paper uses immediate regret as a performance metric and briefly explains what it is. However, it would be better to add a few words about why immediate regret is considered over other metrics, such as simple regret or cumulative regret.
- The authors mention that a different acquisition function was used for LFBO in the experiments, but there is no explanation of why this was done and what acquisition function was used.
- While the paper mentions their proposed method is supposed to be more efficient than the standard PI optimization methods, it would still be helpful to add the standard PI optimization methods as baselines in the experiments.
- The plots in Figure 1 and Figure 2 lack clarity regarding how the results are aggregated and the variability across different runs, e.g., whether they are showing the mean with standard deviation/error bars or the median with other quartiles.
- More thorough experiments can be added. For example, higher dimensions, additional benchmarks, real-world problems.

---

### Official Review · Reviewer_E3hD · 2024-10-04
**Peer review for Lightspeed Black-box Bayesian Optimization via Local Score Matching**

**Rating:** 6
**Confidence:** 3

**Review:**

## Summary

This paper presents a method for improving Bayesian Optimization (BO) by addressing computational complexity and scalability issues associated with high-dimensional black-box function evaluations. The authors propose a fast acquisition function maximization procedure using local score matching to estimate the gradient of the Probability Improvement (PI) acquisition function. The approach allows for gradient-based optimization without the computational burden of fitting typical neural networks. The effectiveness of this approach is supported by theoretical developments and proven by empirical evaluations.

The originality and significance of the paper are present. However, it is recommended the authors to strengthen the result discussion section, which further sheds light on the strengths and limitations of the proposed approach and paves the road for future research.

## Big-picture comments
* **Originality**: the originality of this paper comes from the use of local score matching for gradient estimation in Bayesian Optimization. While the paper specifically focuses on PI acquisition function, the authors pointed out potential future direction to expand into Covariance Matrix Adaptation - Evolutionary Strategy (CMA-ES).
* **Research set-up**: the paper provides both a theoretical foundation and empirical evidence for local score matching method. The paper should, however, include more detailed discussion on the conditions under which the method performs to ensure reproducibility.
* **Result discussion**: The paper could benefit from a more detailed discussion of the results. Particularly, in figure 1, score matching (SM) shows a more promising results than LFBO, compared to figure 2. How should this result be interpreted and what strengths and limitations does it reveal?

## Questions
1. Where do the authors see the local score matching approach the most effective?
2. How should the results in figure 1 and figure 2 be interpreted?
3. What challenges do authors foresee in adapting this approach to other acquisition functions beyond the PI?

---

### Decision · Program_Chairs · 2024-10-09

Accept (Poster)